

# New eastern China agricultural burning fire emission inventory and trends analysis from combined geostationary (Himawari-8) and polar-orbiting (VIIRS-IM) fire radiative power products

Tianran Zhang[1,2], Mark C. de Jong[1,2], Martin J. Wooster[1,2], Weidong Xu[1,2], Lili Wang[3]

[1] King's College London, Department of Geography, Strand, London WC2R 2LS.

[2] NERC National Centre for Earth Observation (NCEO)

[3] LAPC, Institute of Atmospheric Physics, Chinese Academy of Sciences, Beijing 100029, PR China

*Correspondence to*: Tianran Zhang (tianran.zhang@kcl.ac.uk)

**Abstract**

Open burning of agricultural crop residues is widespread across eastern China, and during certain post-harvest periods this activity is believed to significantly influence air quality. However, the exact contribution of crop residue burning to major air quality exceedances and air quality episodes has proven difficult to quantify. Whilst highly successful in many regions, in areas dominated by agricultural burning MODIS-based fire emissions inventories such as GFAS and GFED are suspected of significantly underestimating the magnitude of biomass burning emissions due to the typically very small, but highly numerous, fires involved that are quite easily missed by coarser spatial resolution remote sensing observations. To address this issue, we here use twice daily fire radiative power (FRP) observations from the 'small fire optimised' VIIRS-IM FRP product, and combine it with fire diurnal cycle information taken from the geostationary Himawari-8 satellite. Using this we generate a unique high spatio-temporal resolution agricultural burning inventory for eastern China for the years 2012-2015, designed to fully take into account small fires well below the MODIS burned area or active fire detection limit, focusing on dry matter burned (DMB) and emissions of $CO_2$, CO, $PM_{2.5}$ and black carbon. We calculate DMB totals 100 to 400% higher than reported by GFAS and GFED4.1s, and quantify interesting spatial and temporal patterns previously un-noted. Wheat residue burning, primarily occurring in May-June, is responsible for more than half of the annual crop residue burning emissions of all species, whilst a secondary peak in autumn (Sept-Oct) is associated with rice and corn residue burning. We further identify a new winter (Nov-Dec) burning season, hypothesised to be caused by delays in burning driven by the stronger implementation of residue burning bans during the autumn post-harvest season. Whilst our emissions estimates are far higher than those of other satellite-based emissions inventories for the region, they are lower than estimates made using traditional 'crop yield-based approaches' (CYBA) by a factor of between 2 and 5x. We believe that this is at least in part caused by outdated and overly high burning ratios being used in the CYBA approach, leading to the





overestimation of DMB. Therefore we conclude that that satellite remote sensing approaches which adequately detect
the presence of agricultural fires are a far better approach to agricultural fire emission estimation.

**Keywords**: Agriculture, Biomass Burning, Active Fire, VIIRS, Air Quality, Fire Emission

**1. INTRODUCTION**
Eastern China (111 - 123 °E, 27 – 40 °N) is home to around one third of the Chinese population and includes the area
of the North China Plain and the Yangtze Plain - two of the largest agricultural zones in China (Fig. 1). Cropland
covers over 1.7 million $km^2$ of eastern China, and the region is responsible for an estimated 25% of China's crop
production, including around 51% of the national rice yield (NBSC, 2012). Large amounts of crop residue (~ 60
Tg/year including stems, stalks, straw etc) results from this agricultural production (Chen et al., 2017; Huang et al.,
2012; Zhang et al., 2015), and the burning of this waste in open fields is widespread across much of eastern China
(Fig. 2).
This biomass burning has both local and regional scale air quality impacts, with emissions of particulate matter (PM)
of particular concern (Bond et al., 2013). The East Asian monsoon system that influences much of mainland China
results in prevailing north-westerly to south-easterly atmospheric transport during winter, which is reversed in the
summer months. Under these influences, the smoke from agricultural residue fires in Eastern China often affects
"mega-cities" like Beijing and Shanghai (Chan & Yao, 2008; Cheng et al., 2013; Du et al., 2011; Li et al., 2010).
Modelling studies show that these agricultural emissions can drive intense regional air pollution episodes; Huang et
al. (2012) suggest that $PM_{10}$ concentrations in some cities could reach 600 µg $m^{-3}$ during such episodes, a level 6×
higher than the WHO 24h-mean $PM_{10}$ air quality guideline for human health (WHO, 2005).
Agricultural burning in eastern China accounts for a significant part of China's total biomass burning emissions
(Streets et al., 2003; Chen et al., 2017), however the specific contribution of crop residue burning to air quality
exceedances in China remains uncertain, partly because there is considerable doubt as to the amount of dry matter
burned (DMB) in crop residue fires. For example, his leads to a ~450 % range in total crop residue burning black
carbon emissions in Asia between different emissions inventories (Streets et al., 2003), while emissions estimates of
gaseous species are similarly varied.
A major source of this uncertainty stems from the hitherto relatively poor ability of earth observation (EO) satellite
instruments to adequately detect biomass burning activity in many agricultural areas due to the small size of the fires
usually found in these areas. Many agricultural fields in eastern China are typically only around 700 $m^2$ in area (NBSC,
2012), and fires ignited to burn across the stubble left in the place after harvest are therefore hard to detect with
moderate spatial resolution burned area (BA) mapping from sensors such as MODIS, and are made even more elusive
by the common farming practice of pilling up residues into an even smaller area before igniting them (Zhang et al.,
2017; 2018). As mostly BA mapping methods require ~ > 20 % of a pixel to be burned in order for it to be classified





as 'fire affected' (Giglio et al., 2006; 2009), BA-based emissions inventories such as GFED tend to significantly
underestimate fire activity in areas such as eastern China (Zhang et al., 2018).
Infrared based Active fire (AF) based detection techniques can discriminate fires covering only 0.01-0.1 % of a pixel
area (Wooster et al., 2005; Schroeder et al., 2014), and as such should in theory be able to capture far more fire activity
in agricultural areas than BA based methods. Nevertheless, due to the extremely small size of agricultural fires in
eastern China, a large proportion of fire activity remains undetected by AF detection algorithms applied to 'moderate'
spatial resolution imagery (from sensors such as MODIS). This limitation is a key source of uncertainty within the
FRP approach, and indeed in fact can lead to biased (underestimated) FRP totals caused by the non-detection of the
lower FRP component of a regions fire regime (e.g. Roberts et al., 2015). Higher spatial resolution polar-orbiting
sensors such as VIIRS can provide the ability to identify an increased number of AFs having lower FRP values,
particularly when used with algorithms optimised for small fire detection (Zhang et al., 2017) (Fig. 2), but they still
only capture fires burning in clear skies at the time of the satellite overpass (Giglio et al., 2003; 2006). This limitation
is also a considerable source of uncertainty, and a hinderance given the sometimes short duration of active burning
(especially of  agricultural fires) and the typical polar orbiting imaging frequency of only a few times per day. To cope
with this issue, FRP-based emissions inventories such as GFAS based upon AF methods are generally required to
make assumptions or exploit additional data on the timing and relative diurnal variability of fire activity occurring
between polar orbiting overpasses in order to estimate, for example, total daily Fire Radiative Energy (FRE) (Kaiser
et al., 2012; Xu et al., 2017; Zhang et al., 2017). Here we provide this additional information by exploiting new fire
diurnal cycle information taken from the geostationary satellite Himawari-8, combining it with twice daily FRP
information provided by the 'small fire optimised' VIIRS-IM product of Zhang et al. (2017) to produce a unique high
spatio-temporal resolution agricultural fire dataset (referred to hereafter as the VIIRS-IM/Him dataset) for eastern
China based on FRE totals. This new inventory is designed to reduce bias and uncertainty caused by use of one FRP
data type alone, and to account for small fires burning even for short periods and often well below the MODIS AF
and BA detection limit. The fuel for these fires is waste straw and other agricultural residues, and we use a crop
rotation map to classify the type of agricultural residue being burned at each observed location and time. It is then
used to select the most appropriate smoke emissions factor for calculating the final fire emissions totals from FRE
derived estimates of dry matter burned (DMB).


**2. DATASETS**
2.1 Polar Orbiting VIIRS-IM FRP Product
The Visible Infrared Imaging Radiometer Suite (VIIRS) instrument is currently flown aboard the polar orbiting Suomi
NPP (since 2011) and NOAA-20 (since 2017) satellites and expands upon the capabilities of the AVHRR and MODIS
instruments for environmental monitoring (Zhou et al., 2019). VIIRS has 22 channels spanning the visible to the
longwave infrared, a 3000 km swath width, and nadir pixel resolution ranging between 375 m and 750 m (Goldberg
et al., 2013). Furthermore, a 'pixel aggregation' scheme is applied to VIIRS which limits pixel area increase with scan
angle to a maximum of 4× compared to MODIS' 10× (Wolfe et al., 2013).
With a necessary emphasis on the detection of small fires typical of agricultural regions, our work focuses on
generating a gridded daily biomass burning fuel consumption product that estimates DMB and emissions from the
VIIRS-IM AF Detection and FRP product developed and optimised for eastern China by Zhang et al. (2017), using
data from the instrument aboard the Suomi NPP satellite with a mean local daytime overpass time of 13:30 in the
ascending node, and a mean local nighttime overpass time of 01:30 in the descending node (Wolfe et al., 2013). Fig.
2 shows an example of the VIIRS-IM FRP product, generated from the two observations per day provided by Suomi
NPP VIIRS. This FRP product blends the advantages of the 'small fire' sensitivity of the VIIRS 375 m I-Band, with
the ability to retrieve fire radiative power (FRP) over larger fires using the 750 m M-Band observations. Due to the
very small size of agricultural fires in China, and because the VIIRS I-Band pixel area is 10× smaller than the pixel
area of MODIS, far more fires can be detected in eastern china using the VIIRS-IM AF product of Zhang et al. (2017)
than can be identified in near simultaneous MODIS data, and on average across eastern China retrieves FRP totals
around 4× higher (Zhang *et al.*, 2017).

2.2 Geostationary Himawari FRP Product
To convert the twice-daily VIIRS-IM FRP product to daily-integrated FRE, information on the fire diurnal cycle is
required (Ellicott et al., 2009; Freeborn et al., 2008; Roberts et al., 2009). We obtained this from 10-min temporal
resolution observations from the geostationary Himawari-8 satellite, whose data have recently been used to derive AF
detections and FRP metrics across Asia by Xu et al. (2017). Himawari cannot be used in isolation to directly estimate
daily FRE for each of the 4-years of the study, because (i) Himawari data are only available from early 2015 onwards,
and (ii) Himawari's relatively coarse pixel size (2 km at the sub-satellite point) means that it omits even more of the
agricultural fires than does MODIS (as illustrated by Xu et al., 2017 and in Fig.3). However, where agricultural fires
are concentrated in sufficient density, observations by Himawari do enable their detection and these data can be used
to map the changing FRP of these fires over the day for derivation of the fire diurnal cycle.

2.3 Crop Rotation Map





The predominant agricultural residues burned across eastern China are wheat, corn and rice straw (Huang et al., 2012).
To classify the likely residue type of each detected fire, a crop rotation map (Fig. S1) was generated from the
MIRCA2000 0.08º global monthly crop area dataset (Portmann *et al.,* 2010), which has a spatial resolution equivalent
to 9.2 km × 9.2 km at the equator. These data were used to assign fire activity to a particular crop residue type, which
determined the appropriate agricultural biomass burning emission factors to apply (see Section 3.3).

2.4 Land Cover Data
We use the GlobeLand30 land cover product (Chen et al, 2015) to classify land cover/use for our study area in Eastern
China. GlobeLand30 provides 30m spatial resolution land cover data for a baseline year of 2010 derived primarily
from Landsat (TM5 & ETM +) and China Environmental Disaster Alleviation Satellite (HJ-1) imagers. Fig. 1 shows
the spatial distribution of the agricultural land ratio (regridded to 0.01 degree spatial resolution) calculated use this
dataset in eastern China.

2.5 GFED & GFAS Emissions Inventory Data
The results from the combined VIIRS-IM and Himawari FRP based emissions (VIIRS-IM/Him) dataset were
compared to two state-of-the-art global fire emission databases, the Global Fire Emissions Database (GFED) and the
Global Fire Assimilation System (GFAS). GFED was built to combine remotely sensed data on BA with fuel loads
from the CASA biogeochemical model of vegetation growth, producing monthly, spatially explicit pyrogenic fuel
consumption, carbon, GHG and air pollution emission estimates at 0.25º grid cell resolution globally (Van der Werf
et al., 2010; Giglio et al., 2013). The most recent version (GFED4.1s) includes a "small fire boost" based on AF
detections, in an attempt to counteract the inability of the MODIS BA product to detect many agricultural fires
(Randerson et al., 2012; Van der Werf et al., 2017). Due to this 'boost' GFED4.1s shows higher values of dry matter
burned (DMB) in most eastern China grid cells compared to the 'unboosted' GFED4, and a more extensive fire
distribution. However, Zhang et al. (2018) show that the boosting procedure can introduce significant anomalies into
the GFED dataset at certain times of year, generated when MODIS' AF detection procedure incorrectly identifies
urban features in eastern China as fires.
In contrast to GFED, the GFAS fire emissions database is based on AF detections and is integrated into Copernicus
Atmosphere Monitoring Service (CAMS) system for near-real-time atmospheric composition monitoring and
forecasting. Developed by Kaiser et al. (2012) and based on the FRP method, MODIS supplies the FRP data for the
current GFAS v1.2 up to 4 times per day at most latitudes. From these observations, DMB is calculated via a regression
against GFED DMB values (Kaiser et al., 2012) and daily emissions of 40 emitted species are then calculated at 0.1º
spatial resolution.





2.6 Crop Yield Based Approach Emissions Inventory Data
The traditional method for estimation of agricultural fire emissions is the so-called crop yield based approach (CYBA),
and we compare data from such approaches to our new VIIRS-IM/Him methodology. CYBAs typically calculate the
amount of crop residue burned in a region using a combination of crop production statistics and related additional
parameters using following equation:
$DMB = \sum_{i=1}^{n} P_i R_i B_i C$     (1)
Where $i$ stands for each of $n$ different crops; $DMB$ is total dry matter burned (kg) in the region; $P_i$ is the regional
production of crop $i$ (kg), and is usually derived from annual agricultural statics reports; $R_i$ is the dry matter production-
to-residue ratio (unitless), which depends on the crop type $i$; $B_i$ is the proportion of residue burned in the field for crop
type $i$ in the region under study (i.e. the 'burning ratio'; 0-1, unitless); and $C$ is crop combustion completeness (0-1,
unitless, Huang *et al.,* 2012). DMB is then multiplied by appropriate particulate/gaseous emission factors in order to
estimate the total emissions from agricultural burning.
Certain of the parameters of Eqn. 1 are not so easily determined. For example, the burning ratio ($B_i$) is often based on
questionnaires or investigations on the use of crop residues conducted with farmers (Gao *et al.,* 2002; Wang and Zhang,
2008). Because of strong variations in socio-economic development across the huge expanse of mainland China, large
differences in the estimates of $B_i$ exist (Jiang *et al.,* 2012; Liu *et al.,* 2008; Yamaji *et al.,* 2010). $B_i$ may also change
considerably from year to year since it is strongly impacted by the level of local economic development, the
availability of alternative uses for crop residues in the region, and the regional governance of fire prohibition (Chen
*et al.,* 2017). Moreover, considering the official prohibition of open air burning, the reliability of data based on surveys
that ask farmer how much residue they burn is questionable. Despite this, most studies that include estimation of
agricultural fire emissions in Eastern China have relied on the CYBA (e.g. Cao *et al.,* 2006; He *et al.,* 2011; Huang *et*
*al.,* 2012; Li *et al.,* 2009; Qin and Xie, 2011; Yan *et al.,* 2006; Zhao *et al.,* 2015).

**3. METHODOLOGY**
3.1 Data Gridding and Cloud Cover Adjustment
The VIIRS-IM FRP product data (in MW), originally derived at the pixel scale, were aggregated to 0.1° resolution for
this analysis. Unlike the daily average MODIS FRP calculation of GFAS, which weights individually contributing
MODIS FRP observations by their view zenith angle to downgrade the importance of far off-nadir measurements
(Kaiser *et al.,* 2012), no such weighting was applied to the VIIRS-IM FRP data since they have already shown very
limited view zenith angle dependence as a result of the VIIRS' pixel-averaging procedure (Zhang et al., 2017). For
each VIIRS overpass, the total observed FRP present in each 0.1° grid cell $j$ (i.e. $FRP_j$) was calculated from the
cumulative FRP of all native resolution AF pixels $i$ within the grid cell:
$FRP_j = \sum_{i \in j} FRP_i$     (2)



Total observed agricultural area ($A$, excluding cloud covered area) within each 0.1° grid cell was calculated similarly
using the GlobeLand30 30m landcover map:
$A_j = \sum_{i \in j} A_i$                                            (3)
The VIIRS-IM product is only affected to a limited degree by smoke because of the relative transparency of smoke
plumes at Mid-Wave Infrared (MWIR)  wavelengths due to the dominant particle size being smaller than the
wavelengths of the VIIRS MWIR channel (Zhang et al., 2017). However, the product cannot provide information in
cloud covered areas, and so an adjustment is required to take into account actively burning fires hidden from view by
clouds. Following Streets *et al.* (2003) we assume that for partially cloud covered grid cells, the AF and FRP
distribution under cloud is the same as under the clear sky areas, as is also assumed in GFAS (Kaiser *et al*., 2012).
Subsequently, the gridded and cloud-adjusted FRP areal density ($\rho_j$, MW.km$^{-2}$) is calculated using:
$\rho_j = \dfrac{FRP_j}{A_j}$                                            (4)

3.2 Diurnal Cycle and Daily FRE Generation
Hourly averages of the 10-minute FRP data from the Himawari-8 FRP product of Xu *et al.* (2017) were gridded to the
same 0.1° grid cell resolution as the VIIRS-IM dataset. For each grid cell and calendar day, hourly FRP data were
normalised in order to minimise the impact of day-to-day variations in fire activity:
$\widetilde{FRP_{j,d}^h} = \dfrac{FRP_{j,d}^h - \min(FRP_{j,d})}{\max(FRP_{j,d}) - \min(FRP_{j,d})}$                                            (5)
Where $\widetilde{FRP_{j,d}^h}$ is the normalised Himawari-8 FRP for hour $h$ on day $d$ for grid cell $j$; $FRP_{j,d}^h$ is the observed Himawari-
8 FRP (MW) for hour $h$ on day $d$ for grid cell $j$; $\max(FRP_{j,d})$ and $\min(FRP_{j,d})$ are respectively the maximum and
minimum hourly Himawari-8 FRP (MW) observed on day $d$ for grid cell $j$. Note that $h$ is in local time (UTC/GMT +
8 hours) and the diurnal cycle runs from 0 to 23 hours.
$\widetilde{FRP_{j,d}^h}$ data for 2015 were used to produce two normalised 'seasonal' diurnal fire cycles for the eastern China study
area: a 'summer' diurnal cycle, constructed from May-June data, and an 'autumn' diurnal cycle, constructed from
Sept-Oct data. Both normalised seasonal diurnal cycles were calculated using a weighted mean so that days and grid
cells with high fire activity had the greatest influence on the cycle:
$$FRP^h = \frac{\sum_d \sum_j \left( \widetilde{FRP_{j,d}^h} \times FRP_{j,d}^h \right)}{\sum_d \sum_j \left( FRP_{j,d}^h \right)}$$                                            (7)

Where $FRP^h$ is the normalised FRP for hour $h$ for the entire study area and fire season (summer or autumn). Fig. 4
shows the resulting weighted mean fire diurnal cycle for the summer season for Eastern China. This diurnal cycle is



bi-modal: a primary peak occurs around 13:00 local time that extends from around 08:00 to 18:00 (daytime) and a
second much smaller peak occurs around 21:00 local time (with a height of only ~ 20% of the normalised FRP value
of the first peak).
We blended information from the Himawari FRP diurnal cycle with the instantaneous twice-daily VIIRS-IM FRP
areal density ($\rho_j$, MW.km$^{-2}$) data, using an approach based on Andela et al. (2015) to create the VIIRS-IM/Him dataset.
Here we represent the diurnal fire cycle as a gaussian function parameterised using the Himawari FRP diurnal cycle,
superimposed on a fixed baseline. For a given grid cell $j$, at instantaneous time $t$, VIIRS-IM/Him FRP areal density is
calculated by:

$$\rho_{VIIRS-Him_{j,t}} = \rho_{VIIRS_{night,j}} + \mu\left(\rho_{VIIRS_{day,j}} - \rho_{VIIRS_{night,j}}\right) e^{-\frac{\left(t - t_{Himpeak}\right)^2}{2\sigma^2}} \tag{8}$$

Where $\rho_{VIIRS-Him_{j,t}}$ is the instantaneous VIIRS-IM/Him FRP areal density (MW.km$^{-2}$) for grid cell $j$ at time $t$;
$\rho_{VIIRS_{night,j}}$ is the night-time (~01:00 LST) VIIRS-IM FRP areal density value (MW.km$^{-2}$) for grid cell $j$; $\rho_{VIIRS_{day,j}}$
is the day time (~13:00 LST) VIIRS-IM FRP areal density value (MW.km$^{-2}$) for grid cell $j$; $\mu$ is an adjustment factor
used to account for the difference between the VIIRS daytime overpass time and the peak time of the weighted mean
fire diurnal cycle (see below); $t_{Himpeak}$ is the time of day at which the seasonal Himawari FRP diurnal cycle peaks; $\sigma$ is
the standard deviation of the main peak of the Himawari FRP diurnal cycle, calculated by fitting a gaussian function
(using non-linear least squares) to the seasonal Himawari FRP diurnal cycles. The summer diurnal cycle $\sigma$ value
(2.39±0.053) was applied during the April-August period, and the autumn diurnal cycle $\sigma$ value (1.63±0.041) was
applied during the September-March period.
The adjustment factor $\mu$ is used to account for the fact that the VIIRS daytime overpass time is unlikely to coincide
with the peak of the fire diurnal cycle:

$$\mu = e^{\frac{\left(t_{VIIRS_{day,j}} - t_{Himpeak}\right)^2}{2\sigma^2}} \tag{9}$$

Where $t_{VIIRS_{day,j}}$ is the local time of the VIIRS-IM FRP observation for grid cell $j$.
Daily FRE was then calculated for each grid cell j and calendar day by integrating the instantaneous VIIRS-IM/Him
FRP data using Eqn. 8.





3.3 Conversion to Dry Matter Burned (DMB) and Smoke Emissions
To convert the calculated FRE areal density to fuel consumption/DMB, we multiplied FRE by the 0.368 (±0.015)
kg.MJ$^{-1}$ factor derived by Wooster *et al.* (2005) from a series of outdoor experimental straw fires, that were very
similar to the Chinese agricultural residue fires used herein (Zhang et al., 2015). To convert the resultant DMB into
smoke emissions, we used the emission factors of wheat and rice derived from *in situ* measurements in agricultural
areas by Zhang et al. (2015) (Table 1). Corn residue was not a fuel type measured during those experiments, and so
for this fuel type (which was only 16-22% of the total agricultural fuel consumption) we used the emissions factors
for agricultural corn fires from Andreae and Merlet (2001), as is used in GFAS (Kaiser *et al.,* 2012) (Table 1). Together
with the crop rotation map (see Section 2.3 and Fig. S1) the EFs from Table 1 enabled us to select the appropriate
emissions factor for use at a particular location and time of year.
Furthermore, a winter burning season was discovered during November and December (see details in Section 5.1)
when no cultivation crop is shown in the MIRCA2000 data in the study region. Analysis in this study shows that
winter fires are likely to result from the combustion of stored residues from the autumn harvest season, therefore all
fire activity in winter was assigned to crop types (and therefore emission factors) using the crop rotation map from the
previous closest month (October) (Fig. S1). This methodological change is accounted for in the data presented in Fig.

266    5.


**4. BIOMASS BURNING AND EMISSIONS RESULTS**
4.1 Temporal and Spatial Distribution of FRE In Eastern China
Fig. 5 shows the time series of daily mean FRE areal density in eastern China from February 2012 to December 2015,
reported at 0.1º grid cell resolution, and broken down into three main crop residue types. A strong seasonal variation
is seen, with peak activity in summer (May-June) associated with wheat residue burning and a smaller secondary peak
in activity occurring in autumn (Sept-Oct) associated with corn and rice residue burning. In fact, the secondary peak
is a combination of several fluctuations lasting from October until December, further discussed in Section 5.1. Over
the whole 4-year period, wheat crop residues contributed 65% of the total FRE, rice residues 18%, and corn residues

276    17%.

A distinct spatial pattern showing two main burning seasons can also been seen when FRE areal density is mapped
(Fig. 6). During the summer burning season (May-June), most fires are located between 32º N - 36 º N, extending from
112º E - 120º E near the coast. In the autumn season (Sept-Oct), less fire activity occurs than in the summer fire season
and it is more evenly distributed across the entire study area, though there is still a focus of fire activity between 32 -
34º N and 112 - 119º E. Moreover, in the southwest of the study area (29 - 32º N and 112 - 114º E) we see a region
that only appears to undergo substantial burning in the autumn. This is located in the centre of Hubei Province, which
contributes around 12% of the total rice yield of the whole of China (NBSC, 2015). This area contributes to between
10 and 18 % (year dependant) of the total autumn burning season FRE.






4.2 DMB Comparisons to GFAS and GFED
The outputs generated by our combined VIIRS and Himawari processing chain were compared to those of GFAS and
GFED4.1s (Fig. 7). Dry matter burned (DMB) was used as the common comparison metric, as this removes differences
arising from the use of different emissions factors within the inventories. Overall, the VIIRS-IM/Him DMB estimates
are around 2× to 5× higher than those reported for corresponding months by GFAS and GFED 4.1s. As detailed in
Zhang et al. (2017) and discussed in Section 2, VIIRS has the ability to detect far smaller (and lower FRP) fires than
MODIS, due to its far smaller pixel size and the fact that the I-band observations also retain their pixel area more
effectively across the swath. Ultimately, this difference results in far higher DMB being obtained by the VIIRS-
IM/Him inventory compared to the MODIS based GFAS and GFED inventories.
During the summer months of May-June, all three inventories (GFAS, GFED and VIIRS-IM/Himawari) show a clear
peak in DMB, but GFAS and VIIRS-IM/Him show a much sharper peak in June, while GFED's summer burning
season extends one month earlier (May) and later (July). This extended summer fire season reported by GFED is likely
the result false fire reporting, discussed at length in Zhang et al (2018). VIIRS-IM/Him shows a June DMB peak
ranging from 3.30 to 11.2 Tg, 2× higher than GFED4.1s (1.89 - 5.34 Tg) and GFAS (2.00 to 4.30 Tg). It should be
remembered that the conversion of daily average FRP to DMB in GFAS is derived via a calibration to GFED4.1s
(Kaiser *et al.*, 2012), so these two emissions databases understandably report similar monthly DMB totals.
For the autumn (Sept-Oct) burning season, the peaks in the GFAS and GFED inventories are much less pronounced
than the summer burning season peaks (Fig. 7). DMB in October ranges from 0.57 - 1.74 Tg for GFED, significantly
higher than the 0.31 - 0.61 Tg reported by GFAS, but far lower than the 1.62 - 3.05 Tg of the VIIRS-IM/Him inventory.
The VIIRS-IM/Him derived DMB estimates for eastern China are thus 2 to 3× higher than GFED4.1s and 5× higher
than GFAS; these represent larger differences than exist for the earlier summer burning season. This indicates that
agricultural fires burning during the autumn fire season may be on average smaller and/or more isolated from other
fires than they are in the summer burning season, and thus are even more likely to be missed by the MODIS AF
detection product (Giglio *et al.,* 2006) and/or the MODIS BA product (Giglio *et al.,* 2013) than they are during other
more intense burning periods.

4.3 Agricultural Fire Emissions Intercomparison
This section presents a comparison of the total annual agricultural fire emissions calculated using the VIIRS-IM/Him
method with other inventories of Chinese agricultural fire emissions in the literature, and against emissions totals from
other sectors to gain a better understanding of the relative importance of agricultural fire emissions. To compare with
other reported agricultural fire emission inventories for China, the DMB estimates produced herein were converted to
fire emissions estimates using the emissions factors and methods described in Section 3.3; these results are summarised
in Fig. 8 and Table 2.



From Fig. 8, it is clear that wheat residue burning is the primary agricultural emission source, accounting for over 50%
of the total emissions released each year (specifically 55-69% of $PM_{2.5}$, 71-81% of BC, 66-77% of $CO_2$, and 69-80%
of CO). Fig. 8 also indicates a considerable reduction in emissions in 2015 compared to previous years, largely
attributable to a reduction in the amount of wheat residue burnt. For example, total $PM_{2.5}$ emissions from agricultural
residue burning in eastern China for 2012-14 cover a relatively narrow range of 107 - 130 Gg (Fig. 8 & Table 2), but
decrease to $67 \pm 24$ Gg in 2015 due to an almost halving of DMB (Fig. 7); similar patterns are observed for BC, $CO_2$,
and CO (Fig.8).
From Table 2, it is apparent that emissions totals calculated using the VIIRS-IM/Him approach are consistently higher
than those reported by GFAS by factor of 1.2-4.2 (species/year dependent). Similarly, VIIRS-IM/Him emissions totals
for $CO_2$ and PM2.5 are greater than those reported by GFED by a factor of 1.1-1.7. In both cases, this can be explained
by the tendency of MODIS to miss activity from small fires compared to VIIRS. VIIRS-IM/Him emissions for CO
and BC in 2015 are lower than those reported for GFED, which can be attributed to differences in the emissions factors
used between the approaches.
Emissions totals calculated using the VIIRS-IM/Him approach are smaller than those estimated by CYBA studies for
the East China/North China Plain regions (Zhang et al., 2008; Huang et al., 2012; Qiu et al., 2016) by a factor of 2-5.
It is possible that the much higher totals estimated from the CYBA based studies maybe due to the use of very high
residue burning ratios ($B_i$ in Eq. 1) for corn and rice in particular. This finding is discussed further in Section 5.
Liu *et al.,* (2015) estimated total emissions in the North China Plain region (a similar area to the study area used in
this paper) using MODIS FRP-based calculations, and assumed a modified Gaussian function for the diurnal cycle to
generate the daily FRE estimates from which emissions were then derived. These estimates are much closer in
magnitude to the equivalent estimates calculated using the VIIRS-IM/Him method than those from the CYBA studies,
however 2013 & 2014 estimates by Liu et al. are consistently lower (by a factor of 0.3-0.9); again, we attribute this
difference to the fact that MODIS based methods capture less fire activity than our VIIRS-IM/Him approach.
Interestingly, Liu *et al.* (2015) estimated far higher emission totals for 2012 compared to 2013 & 2014 and report
greater total CO and BC emissions than we do. For example, annual $CO_2$ emissions in 2012 (26,000 Gg) are $> 2\times$ their
reported total emissions for 2013 (9800 Gg) and 2014 (13,000 Gg). However, Liu *et al.*'s processing approach did not
provide any adjustment for the impact of the MODIS 'bow-tie' scan geometry effect, which leads to duplicated AF
detections and this FRP towards the edge of the MODIS swath, and which was highlighted as significant issue for
FRP quantification by Freeborn *et al.* (2008) and Zhang et al. (2017). This is a particular problem in MODIS data
from the year 2012, where large amount of duplicated observations have been found towards edge of swath (Fig. S2).
This problem has been addressed in GFAS using a scan-angle dependent weighing factor for the MODIS FRP data
(Kaiser *et al.,* 2012), as described in Section 2.5, and GFAS' $CO_2$ emissions from 2012 are only 24% and 10% higher
than from 2013 and 2014 respectively, a much more modest increase compared to that reported in Liu *et al.* (2015).
Fig. 9 presents a comparison of agricultural emissions calculated using the VIIRS-IM/Him method with emissions
from non-biomass burning sources produced by Li et al. (2014) for a sub-area of eastern China (32-36° N, 112-122°





E) for the year 2013. We note that crop burning emissions are of relatively little significance when considered on an
annual basis; for all four species ($CO_2$, $CO$, $PM_{2.5}$, $BC$), contributions from agricultural residue burning range between
0.56% and 2.0% of total annual emissions, with the majority of emissions resulting from industry and residential
sources. However, in June when agricultural burning and emissions are at a maximum, residue burning contributes
8.1%, 18%, 22% and 20% of total monthly emissions for $CO_2$, $CO$, $PM_{2.5}$ and $BC$ respectively, highlighting the strong
seasonal impact agricultural burning can have on the emission of species that affect both climate and air quality.

**5. ANALYSIS AND DISCUSSION**
5.1 Importance of Wheat Residue Burning
Findings in Section 4 (Fig. 5 & 8) indicate that a larger proportion of wheat residue than corn or rice residue is burnt,
for several reasons. First, the yields of these three crop types in Eastern China are relatively similar - in 2015 for
example, wheat yield was 10% lower than rice yield, and only 20% higher than corn (Table S1; NBSC, 2015). Second,
the dry matter production-to-residue ratio ($R_i$ in Eqn. 1) of wheat is not higher than that of rice or corn (Table S2;
Wang and Zhang, 2008). Third, with the exception of black carbon, the emission factors for wheat residues are broadly
similar to or smaller than the corresponding rice and corn emission factors. It is unknown why a greater fraction of
wheat residue than corn and rice residue is burnt, however, it is possible that local management practices and/or
stakeholder priorities differ depending upon the residue type and time of year at which crops are harvested, ultimately
impacting the fate of these residues e.g. residues from certain crops maybe valuable as fertiliser (Huang et al., 2012),
animal feed or for domestic/local energy production (Chen et al., 2017; Liu et al., 2008).

5.2 Discovery of A Winter Burning Season
As detailed in Section 4.1, small peaks in our dry matter burned (DMB) time-series are apparent in November-
December of each year (grey shaded area shown in Fig. 5). Since no mention of such a winter burning season was
found in the literature (e.g. Chen *et al.,* 2017; Huang *et al.,* 2012; Zhang *et al.,* 2008), these winter peaks were initially
considered to be erroneous and likely caused by VIIRS AF false alarms that had failed to be excluded by the landcover
and/or persistent thermal anomaly masking detailed in Zhang et al., (2017). Furthermore, according to the crop rotation
map derived from the MIRCA2000 data (Fig. S1), there is no obvious harvesting of wheat, corn, or rice during the
winter in eastern China. However, close examination of the original VIIRS data and the VIIRS-IM FRP product
generated from it by Zhang et al., (2017) shows that most of the AF pixels detected in eastern China in winter are in
fact located in or very close to areas classified as agricultural land (Fig. S3), and are not located close to industrial
areas of the type known to cause false AF detections (Zhang et al., 2017), nor do the AF detections appear multiple
times in the same month at the same location, as would be expected if they were false alarms generated by non-fire
features. It therefore seems highly probable that these AF detections are actually a consequence of true agricultural
burning (Fig. S3-5).



The most reasonable explanation for the winter AFs appears to be that some of the crop residues from the Sept-Oct
(Autumn) harvest season were left idle for a few months and burned in the winter, rather than immediately. Local
newspapers, online media and other information sources were consulted, and were found to support the existence of
winter residue burning episodes. One example is a report by Jiangsu Province TV station in 5 December 2013, where
a huge crop residue burning episode was reported in Hongze (Jiangsu Province), close to the location shown in Fig.
S3. Stills from this TV report show flames, thick smoke and extremely poor visibility resulting from the crop residue
burning, described in Chinese language subtitles (Fig. S4). Reports of similar episodes were found in different
websites/newspapers from across much of eastern China (e.g. Wang and Zhang, 2016; Za, 2015; Zuo, 2015).
Subsequent to this confirmation, an explanation as to why this activity may have occurred outside of the normal
burning season was sought. According to Yun Xia, a local governor of the Environmental Department in Hefei
(interview conducted by Anhui News; Zuo, 2015), the prohibition on agricultural burning started at beginning of
September in that area, and continued up until the 20th November. During this period, the local government strongly
enforced its polices aiming to restrict agricultural residue burning, and established almost continuous patrols to
identify areas likely to host crop residue fires in order to prevent their ignition. However, without a widespread and
cost-effective alternative way to dispose of their crop residues, local farmers may simply have stored the residue
material and burned it soon after the end of the prohibition period, when the intensive patrol period had ceased. The
end of the prohibition period coincides almost exactly with the time of the new winter burning season identified by
our VIIRS-IM/Him dataset (Figs. 5- 7).
The winter season is important for biomass burning in this area of China, accounting for between 19 and 36 % (year
dependant) of the combined autumn and winter FRE total. Based on the crop rotation map (Fig S1), this fire activity
was assigned to the burning of both corn and rice residues, with the contribution of each residue to total FRE (and
thus DMB) almost equal (49 % and 51 %, average over all years). This split by residue type is very similar to that
observed in the Autumn burning season (corn = 54 %, rice = 46 %, average over all years), despite the observed
variation in the spatial distribution of fire between autumn and winter (Fig. 6). In general, winter burning appears to
take place closer to provincial capitals than autumn burning does; the reason for this spatial shift in fire is discussed
in Section 5.4.

5.3 Disagreement Between Satellite Derived Emissions and Crop Yield Based Approaches
In Section 4.3, it was noted that annual emissions totals calculated using crop yield based approaches (CYBAs) are
greater than those calculated using the VIIRS-IM/Him method by a factor of 2-3, depending on species. We believe
that this discrepancy relates to the 'burning ratio' (BR) used in CYBA to produce emissions estimates. The burning
ratio is the ratio of crop residue burned in the field compared to the total amount of residue produced by harvesting,
and is a key parameter in bottom up CYBAs (see Eqn. 1, and Chen *et al.,* 2017; Gao *et al.,* 2002; Huang *et al.,* 2012;
Li *et al.,* 2016). Streets *et al.* (2003) used a uniform BR of 17 % derived from 1970's data, however more recent
studies often make use of regionally varying fractions. We identified three sources of regionally varying burning ratios
that are widely used in the CYBA literature:





i)    Wang and Zhang (2008), divided all provinces in China into six zones according to their geographical
distribution. A questionnaire-based survey conducted amongst farmers within these regions was used to
elucidate the level of burning activity, and using the responses it was determined that burning ratios for
the different categories ranged from 11% to 33%. Outputs were applied and referenced in a series of fire
emission studies (He *et al.,* 2011, Qin and Xie 2011, Zhang *et al.,* 2016).

ii)   Gao *et al.* (2002) derived a set of province-dependent burning ratios adopted from a large-scale
investigation of crop residue use across different Chinese provinces. These ratios have been used and
referenced in Huang *et al.* (2012), Yan *et al.* (2006), Zhang *et al.* (2008), and are shown in Fig. 10.

iii)  A derived value based on farmers' income levels, based on the fact that Cao *et al.,* (2006) found a positive
linear correlation between the income of farmers and burning ratio (r = 0.81). This relationship has been
applied within several fire emission studies (Sun *et al.,* 2016, Zhao *et al.,* 2015) and will be examined in
Section 5.4.

Using crop yield information and the DMB data derived from the VIIRS-IM/Him processing performed herein, it is
straight forward to reverse the CYBA methodology to calculate the burning ratio for each crop type. This procedure
can help confirm whether the outputs derived herein are comparable with those of the existing literature, as well as
enabling the advantages offered by the remote sensing time series to be fully exploited. The burning ratios ($B_{ij}$) for
each province *i* and crop type *j* are calculated from:
$$B_{ij} = \frac{DMB_{ij}}{P_{ij}R_iC} \qquad (10)$$
Where $DMB_{ij}$ is the estimated VIIRS DMB (g/m$^2$) for province *j* and crop *i*; $P_{ij}$ is the yield of crop *i* for province *j* (kg);
$R_i$ is the dry matter production-to-residue ratio for crop *i* (unitless) and *C* is crop combustion completeness (proportion,
0-1). The province level crop yield $P_{ij}$ is derived from annually published statistical reports, and are presented in Table
S1. $R_i$ and *C* are from Huang et al., (2012); and are presented in Table S2.
The crop and province dependent burning ratios calculated from the VIIRS-IM/Him data are shown in Fig. 10,
alongside the burning ratios from Gao *et al.* (2002). Fig. 10 indicates that there is considerable variation in burning
ratios between individual provinces, and that VIIRS-IM/Him wheat burning ratios for are clearly much higher than
rice/corn burning ratios. When averaged over the entire Eastern China study area, yearly mean burning ratios from
our results for wheat are highest (7.8 - 12%), followed by corn (1.7 - 2.3%), then rice (0.9 - 2.0%). Equivalent mean
burning ratios calculated using data from Gao *et al.* (2002) are 9.8 %, 5.9 % and 8.5 %, respectively. While VIIRS-
IM/Him wheat residue burning ratios are in reasonable agreement with those used in the various CYBA studies, our
rice and corn burning ratios are much lower; this appears to explain why total annual emissions from the VIIRS-
IM/Him approach are much lower than the total emissions obtained from the CYBA studies.
Fig. 10 also indicates that burning ratios are not only influenced by crop type and province, but also vary considerably
from year to year. For example, in 2012, satellite derived wheat burning ratios for the important agricultural provinces
of Anhui (30%), Shandong (11%), Jiangsu (24%) and Henan (11%) are not dissimilar to corresponding ratios (20%,



8%, 10%, 7% respectively) from Gao *et al.,* (2002). However, during 2015, values derived in this study are much
lower (Anhui = 6 %; Shandong = 4 %; Jiangsu = 4 %; Henan = 6 %). This interannual variation may be linked with
changing local farming activity and prohibition policies (Chen *et al.,* 2017, Li *et al.,* 2016, Yang *et al.,* 2008).
We believe that the disagreement between the burning ratios derived here and those used in CYBA derived studies
indicate that emissions inventories derived using traditional CYBAs may be overestimating agricultural burning
emissions, for two main reasons: (1) there appears to be considerable uncertainty and subjectivity associated with the
methods used to estimating burning ratios used in CYBA studies, and (2) many burning ratios used in CYBA studies
are taken from relatively old (>5-10 years) sources of data. For example, Street *et al.* (2003) use data from 1970's,
while most later researchers use burning ratios from Wang and Zhang (2008) and Gao *et al*. (2002) as listed above in
this section.
As shown by this analysis, burning ratios appear to be subject to high spatial and interannual variability due to rapidly
changing agricultural policies and decision making that influences the fate of crop residues. As such, in order to ensure
reliable emissions estimates, we suggest that future agricultural emission studies and inventories that are based upon
CYBAs should endeavour to use burning ratios derived from data (1) with high granularity, and (2) that was collected
in the corresponding inventory year.

5.4 Influence of Social Factors on Agricultural Burning
As highlighted in Section 5.2, some studies assume a positive relationship between burning ratio and the mean local
income of farmers (Cao *et al.,* 2006; Qin and Xie, 2011). The explanation for this is that higher income areas have
better access to electricity and other energy sources, and thus have less need to utilise crop residues for heating and
cooking – leading to higher ratios of open burning at these locations. However, this is not what we observe in from
analyses carried out for this study. In Fig. 11a, minimal correlation was found between GDP and burning ratio, and
there is some suggestion of an inverse relationship between these variables ($y=-89x+9542$, $r^2=0.13$). When directly
comparing GDP with DMB, as Fig. 12 demonstrates, the provinces with the highest average annual DMB per m$^2$
(Anhui and Henan; 46 and 27 g.m$^{-2}$.yr$^{-1}$ respectively) have lower GDP values (US$ 5,580 and 5,335 per capita) than
provinces with lower annual DMB densities (e.g. Shandong and Jiangsu, with 15 and 21 g.m$^{-2}$.yr$^{-1}$ respectively) but
high GDP per capita (USD$ 9,882 and 13,311 respectively). In fact, across the eastern China study area, our annual
total DMB metric was found to be somewhat inversely correlated with GDP per capita ($r^2 = 0.33$; Fig. 11b).
We theorise that the observed inverse correlation between GDP and DMB results from the fact that alternative residue
disposal methods to biomass burning have a relatively high cost, and can only be afforded by wealthier
farmers/provinces. For example, the local government of Jiangsu Province (a relatively wealthy province [$ 13,311
per capita] with only moderate DMB [21 g.m$^{-2}$.yr$^{-1}$]) released a regulation in 2009 stating that by the end of 2012, over
35% of crop residues should been incorporated into the soil after mechanised harvesting. The regulation also indicated
that the local government should include a budget for improving the efficiency of agricultural machinery and subsidise
farmers who follow this regulation. Furthermore, alternative uses for crop residues are often expensive, and are likely





only a viable option in relatively wealthy areas. For example, research on residue burning for power generation shows
the government needs to pay at least 20% of the total cost of the operation to keep the power plants running, partly
because of the high costs associated with residue collection and transportation from the fields (Li and Hu, 2009).
In addition to influencing the quantity of material burned and when it is burned, societal factors also appear influence
the spatial pattern of burning within provinces, and at more granular levels such as at the 0.1° grid cell level. The work
presented in Section 5.1 suggests that the winter burning season (Nov-Dec) is caused by delayed burning of residues
left over from the autumn harvest season, because of prohibition policies related to burning being more robustly
enforced earlier in the season. Fig. 6 also showed that the spatial distribution of FRE areal density during winter is
different from the normal autumn burning season that occurs in Sep-Oct. Generally, the areas of strongest burning are
further from the provincial capital cities (marked by the green stars in Fig. 6) during autumn. For example, fires in
Anhui Province are mainly distributed in the north during autumn, whilst fire locations change to the south (closer to
the capital city of Hefei) during the delayed winter burns. A similar example can also be seen in Hubei Province,
where fires shift from west to east from the autumn to winter burning seasons.
To examine this in a more quantitative manner, we calculated the distance from each grid cell shown in Fig. 6 to their
provincial capitals. Fig. 13 shows the normalised frequency distribution of the distance from the capital to the top 10%
of FRE releasing grid cells in each province, using data from the four burning seasons during the 2012-2015 period.
The first and third distance quartiles during the autumn season are 109 km and 214 km respectively, but for the 'lagged'
winter burning season, the distribution shifts to far shorter distances (first and third quartiles of 70 km and 153 km
respectively). Similarly, the mean distance from provincial capitals also decreased from 165 km in autumn to 124 km
in winter. A Kolmogorov–Smirnov (K-S) test was performed to evaluate the difference between the distributions of
distance data for the autumn and winter burning seasons, and the resulting high K-S statistic ($0.30, p < 0.001$) indicates
that the distribution of distances during the winter months is substantially different to the autumn distance distribution.
Similar results were found when we applied the K-S test to each calendar year of data separately (not shown). One
possible explanation for this observed difference is that the geographical shift might also be linked with the policies
aimed at prohibiting burning, since areas close to capital cities are likely to have more resources for enforcing the
prohibition compared to areas more distant from the major urban populations.


**6. SUMMARY AND CONCLUSION**
We have developed a new state-of-the-art agricultural burning emissions inventory ('VIIRS-IM/Him') for eastern
China by combining fire radiative power (FRP) observations from the VIIRS and Himawari-8 sensors for the 2012-
2015 period. While several other studies have also used satellite EO data to develop such inventories, they have all
relied on MODIS fire products for their source observations. Such inventories include the global GFED and GFAS
inventories, several Chinese regional studies (e.g. Huang et al., 2012, Liu et al., 2015). MODIS fire products are
known to show very high omission rates in environments dominated by small agricultural fires (Randerson et al., 2012;





Zhang et al., 2017, 2018), but the 'small fire optimised' VIIRS-IM product of Zhang et al. (2017) used in this study
detects far more of the fire activity across eastern China and on average show FRP totals around 4x higher than those
of the MODIS AF products. To convert the twice-daily VIIRS-IM FRP product information to daily time-integrated
FRE, we have used new diurnal fire cycle data from Himawari-8, a geostationary satellite positioned over east Asia
that can best capture the specific diurnal fire variability of the agricultural burning regions.
Our final VIIRS/Him agricultural fire emissions inventory reports dry matter burned (DMB) totals around 2-5× higher
than is reported by GFAS and GFED 4.1s in eastern China for corresponding time periods. Use of a crop rotation map
allowed our VIIRS-IM/Him fire and emissions outputs to be disaggregated by individual crop types, and we found
wheat residue burning to be the primary agricultural emission source, accounting for over 50% of the total emissions
each year for all investigated smoke constituents ($CO_2$, CO, $PM_{2.5}$ and black carbon). A strong seasonal variation in
fire activity and emissions is seen, with annual peak activity occurring in summer (May-June) as a result of wheat
residue burning, and a smaller secondary activity peak occurring in autumn (Sept-Oct) as a result of corn and rice
residue burning. Furthermore, we discovered a new winter (Nov-Dec) agricultural residue burning season. As no crop
harvesting occurs during winter, we suspect that this fire activity results from farmers burning previously stored
residues from the autumn harvest in winter, after autumn residue burning prohibitions have been lifted. This theory is
supported by our observation of statistically distinct spatial burning patterns in the autumn and winter seasons; the
majority of autumn burning occurs at a greater distance from provincial capitals than the winter burning does. This
may reflect stronger enforcement of autumn residue burning prohibition measures in close proximity to major urban
population centres than in rural locations. Farmers in areas with stronger prohibition enforcement (typically closer to
urban areas) then burn their agricultural residue in winter.
Detailed comparison to existing inventories showed that our VIIRS-IM/Him annual emissions totals are 1.2-4.7×
greater than those reported by GFAS, and 0.5-1.7x those reported by GFED4.1s, with some inter-species variability
due to the use of different emissions factors between the inventories. By contrast, the VIIRS-IM/Him inventory shows
emissions totals that are on average lower than those from emission inventories derived using crop yield based
approaches (CYBA) by a factor of 2-5x. This discrepancy is believed to be primarily due to many CYBAs using
outdated and/or inappropriate burning ratios, that consequently leads to CYBAs overestimating the amount of crop
residue DMB annually. Back calculated burning ratios from the VIIRS-IM/Him data suggest that burning ratios for
rice and corn are much lower than the CYBA literature suggests (approx. 0.9-2.3 % rather than 11-33 %). We also
noted considerable inter-provincial and interannual variation in these back calculated burning ratios, for example,
wheat burning ratios significantly decrease over our four-year study period. This strongly suggests that high spatial
resolution, up-to-date burning ratios should always be used in CYBA for agricultural burning fire emission estimation.
Furthermore, several CYBA approaches (e.g. Sun *et al.,* 2016, Zhao *et al.,* 2015) have derived burning ratios from
provincial GDP data, assuming a positive relationship between these variables (Cao *et al.,* 2006). However, we found
evidence of an opposite (i.e. negative) relationship between provincial GDP and the amount of DMB in agricultural
fires, hypothesised to be due to the higher cost of disposal of crop residues by non-biomass burning methods. This
suggests that great care needs to be taken when deriving burning ratios for use in future agricultural emissions





inventories based upon CYBA methods, and that satellite remote sensing approaches based on EO datasets that
adequately detect the presence of agricultural fires are a far better approach to fire emissions estimation in such
environments.


**ACKNOWLEDGEMENTS**
This work has been supported by the NERC National Centre for Earth Observation (NE/R000115/1) and specifically
by NERC Grant NE/M017729/1. The VIIRS SDR and MODIS data were retrieved from CLASS and Reverb, and are
courtesy of the NASA EOSDIS LP DAAC and USGS EROS Centre, South Dakota. GFAS data was generated using
Copernicus Atmosphere Monitoring Service Information, operated by ECMWF. All data storage and processing were
conducted using the UK's JASMIN super-data-cluster system, managed by UK STFC's Centre for Environmental Data
Analysis (CEDA).

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

**Table 1:** Emission Factors for agricultural residue burning used in this study. Wheat and rice emission factors were
derived from field measurements conducted in eastern China and reported by Zhang et al. (2015), while the corn
emission factors are from Andreae and Merlet (2001), the same as those used in GFAS (Kaiser *et al.,* 2012). *$PM_{2.5}$
= particulate matter with diameter < 2.5μm

|  | Emissions Factor (g.kg$^{-1}$) | | |
|---|---|---|---|
|  | Wheat | Corn | Rice |
| $CO_2$ | 1739±19 | 1308±14 | 1761±30 |
| CO | 60±12 | 92±18 | 47±19 |
| $PM_{2.5}$* | 6.1±1.3 | 8.3±1.8 | 9.6±4.3 |
| Black Carbon | 0.70±0.09 | 0.42±0.05 | 0.56±0.04 |

**Table 2:** Total species-specific fire emissions calculated in this study for agricultural burning in eastern China, and
comparison to those contained within other fire emissions inventories and calculated in previous studies.

| Reference | Region | Year | Method | Emissions (Gg.yr$^{-1}$) | | | |
|---|---|---|---|---|---|---|---|
|  |  |  |  | $CO_2$ | CO | $PM_{2.5}$ | BC |
| This study | Eastern China | 2012 | Satellite | 31066 ± 1960 | 1035±327 | 124±43 | 11±1.8 |
|  |  | 2013 |  | 31107 ± 1748 | 1025±320 | 130±44 | 11±1.7 |
|  |  | 2014 |  | 27069 ± 1421 | 904±279 | 107±36 | 10±1.5 |
|  |  | 2015 |  | 16932 ± 1044 | 562±177 | 70±24 | 6±0.95 |
| GFAS | Eastern China | 2012 | Satellite | 9219 | 649 | 58 | 3.0 |
| Kaiser *et al.,* 2012 |  | 2013 |  | 8173 | 576 | 52 | 2.6 |
|  |  | 2014 |  | 8760 | 617 | 55 | 2.8 |
|  |  | 2015 |  | 6818 | 480 | 43 | 2.2 |
| GFED4.1s | Eastern China | 2012 | Satellite | 18629 | 1199 | 74 | 8.8 |
| Van der Werf *et al*., 2017 |  | 2013 |  | 24034 | 1547 | 95 | 11 |
|  |  | 2014 |  | 18241 | 1173 | 72 | 8.6 |



| | | | | | | | |
|---|---|---|---|---|---|---|---|
| | | 2015 | | | 15892 | 1023 | 63 | 7.5 |
| Liu *et al.,* 2015 | NCP[1] | 2012 | Satellite | 26000 | 1700 | 102 | 13 |
| | | 2013 | | 9800 | 630 | 39 | 5 |
| | | 2014 | | 13000 | 820 | 50 | 6 |
| Zhang et al., 2008 | Eastern China[3] | 2004 | CYBA[2] | 67703 | 5624 | - | - |
| Huang et al., 2012 | Eastern China[3] | 2006 | CYBA | 41374 | 2668 | 164 | 20 |
| Qiu et al., 2016 | Eastern China | 2013 | CYBA | 72071 | 2549 | 445 | 42 |
| Li et al., 2016 | NCP | 2012 | CYBA | 68675 | 5983 | 452 | 23 |
| Sun et al., 2016 | China | 2013 | CYBA | 192540 | - | - | - |
| Street *et al.,* 2003 | China | 2000 | CYBA | 160000 | 10000 | - | 70 |
| Yan *et al.,* 2006 | China | 2000 | CYBA | 184000 | 11000 | 470 | 80 |

[1] NCP refers to the North China Plain, which has a geographic extent similar to that of this study (32-41ºN, 113-121ºE).
[2] CYBA refers to Crop Yield Based Approaches, see Section 2.6.1
[3] Sum of provinces/cities shown in Fig.1 of this study.

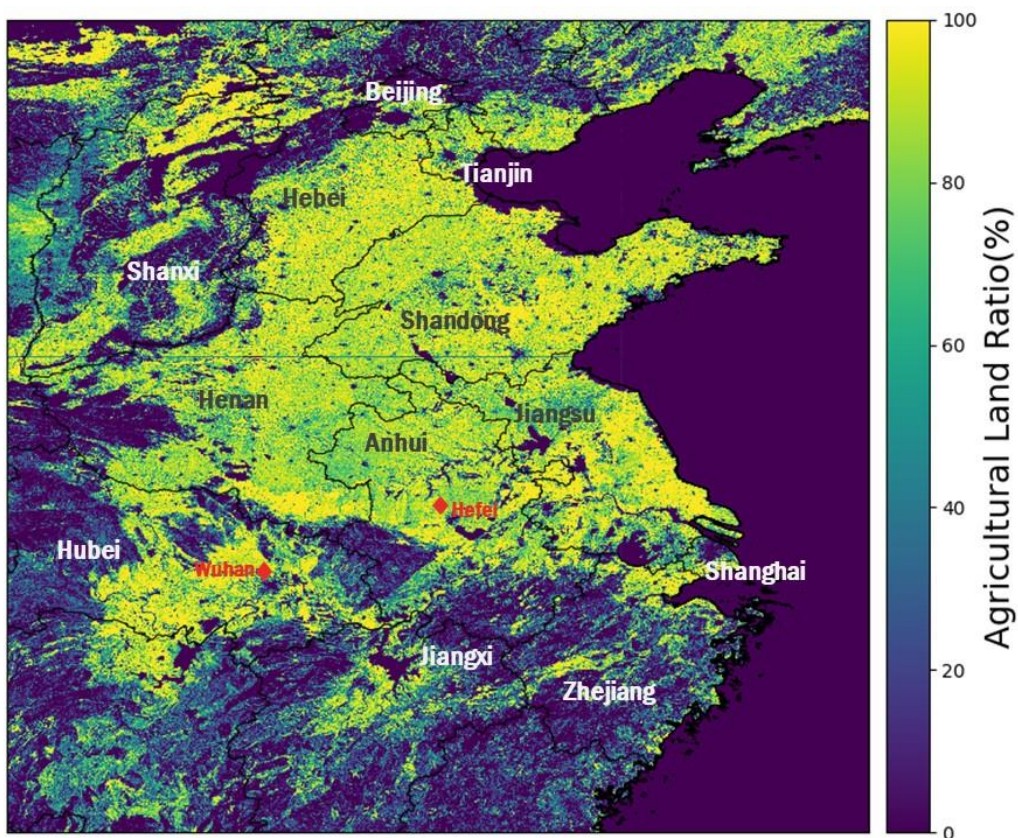

**Figure 1:** The spatial extent of the study area (111-123° E, 27-40° N). The agricultural land ratio taken from the GlobeLand30 land cover product (Chen et al, 2015) was re-gridded to 0.01 degree spatial resolution, and is overlain with the main provinces, mega-cities and some important provincial capital cities in eastern China. The basic layer of country/province borders within this map was created using Python Basemap librabry.

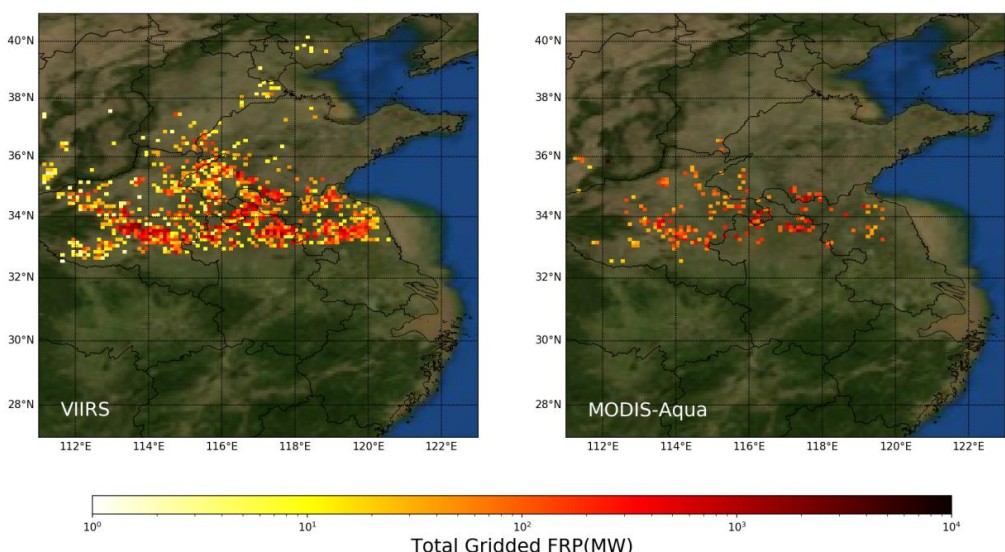

**Figure 2:** Example of the spatial distribution of total gridded FRP (MW; calculated per 0.1° grid cell) calculated from near simultaneous VIIRS-IM and MODIS Aqua data collected over the eastern China study area of Fig. 1 on June 12th, 2012. The VIIRS-IM data product clearly quantifies a higher proportion of the FRP from fires burning in the region at the time of the satellite overpass than MODIS Aqua does. The basic layer of country/province borders within this map was created using Python Basemap librabry.

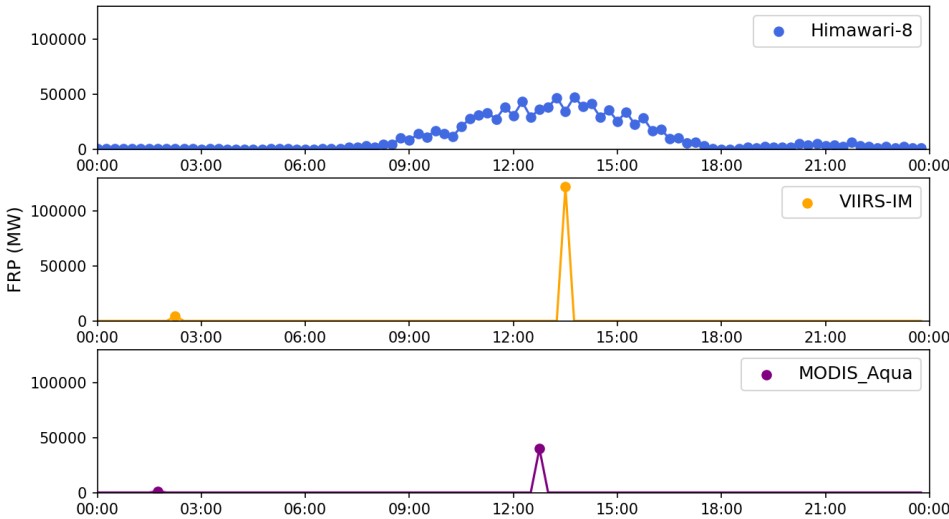

**Figure 3:** Time series of spatially summed FRP for eastern China, as retrieved from geostationary Himawari, and polar-orbiting VIIRS-IM and MODIS observations made on June 11th, 2015. VIIRS and MODIS Aqua provide





typically two observations per day, and sometimes three when the swath overlaps from different orbits occur.
Himawari provides 144 observations per day.

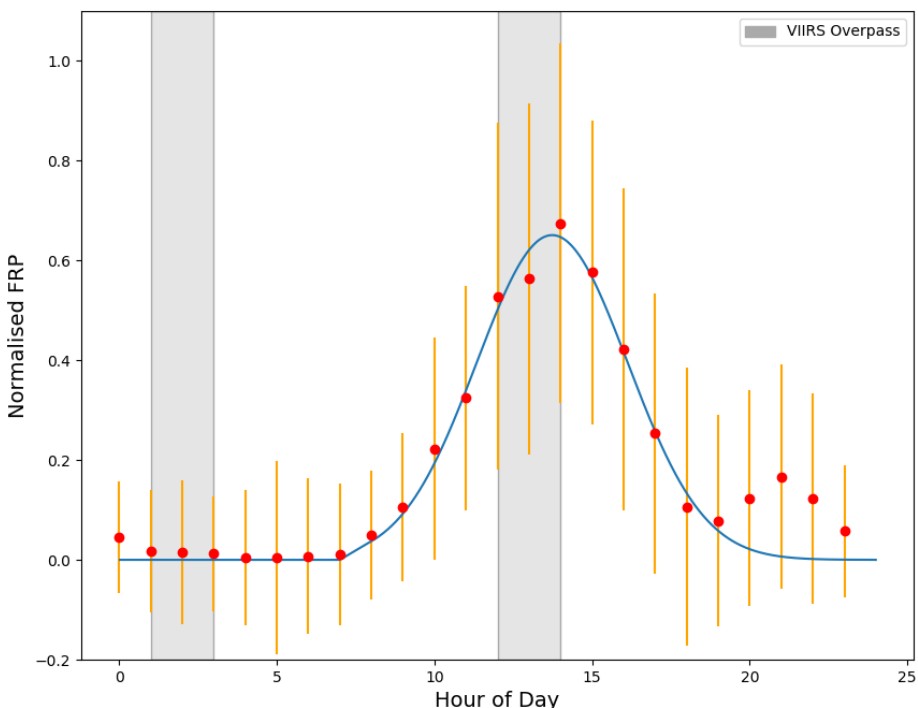

**Figure 4:** Time series of hourly normalised fire radiative power derived from Himawari-8 FRP data generated using
the algorithm of Xu et al. (2017) over eastern China at 0.1 degree for June 2015 (the 'Summer' diurnal fire cycle).
The blue curve shows the best fit of the Gaussian distribution, with orange error bar show standard deviation. Grey
shading shows the two daily VIIRS overpass periods.

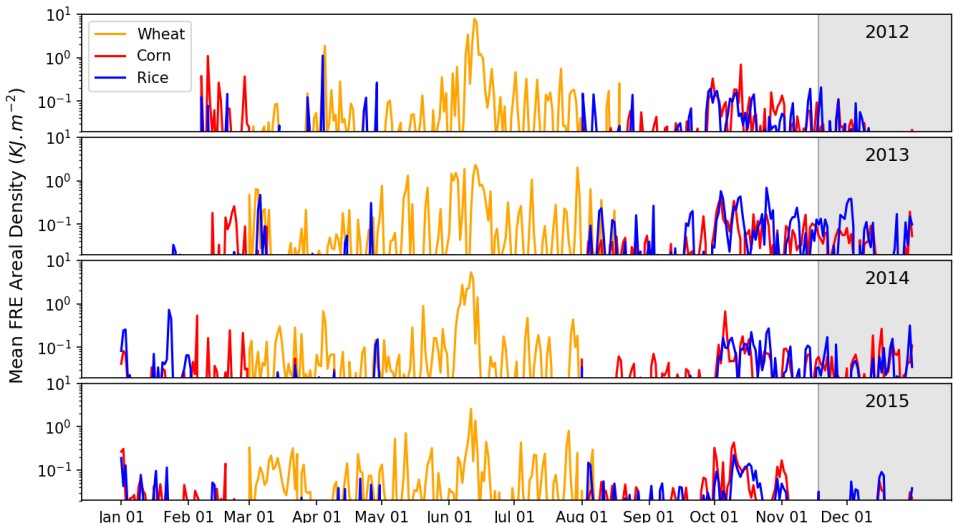

**Figure 5:** Time-series of mean daily FRE areal density (kJ m⁻², calculated per 0.1° grid cell) from 2012-2015 for the entire study area disaggregated by crop residue type (wheat, corn and rice) according to the method described in Section 2.4. Grey shaded areas highlighted the usual newly discovered winter burning season from mid-November to December when no crop harvesting occurs but where fires are clearly occurring. This period of agricultural burning is discussed further in Section 5.1

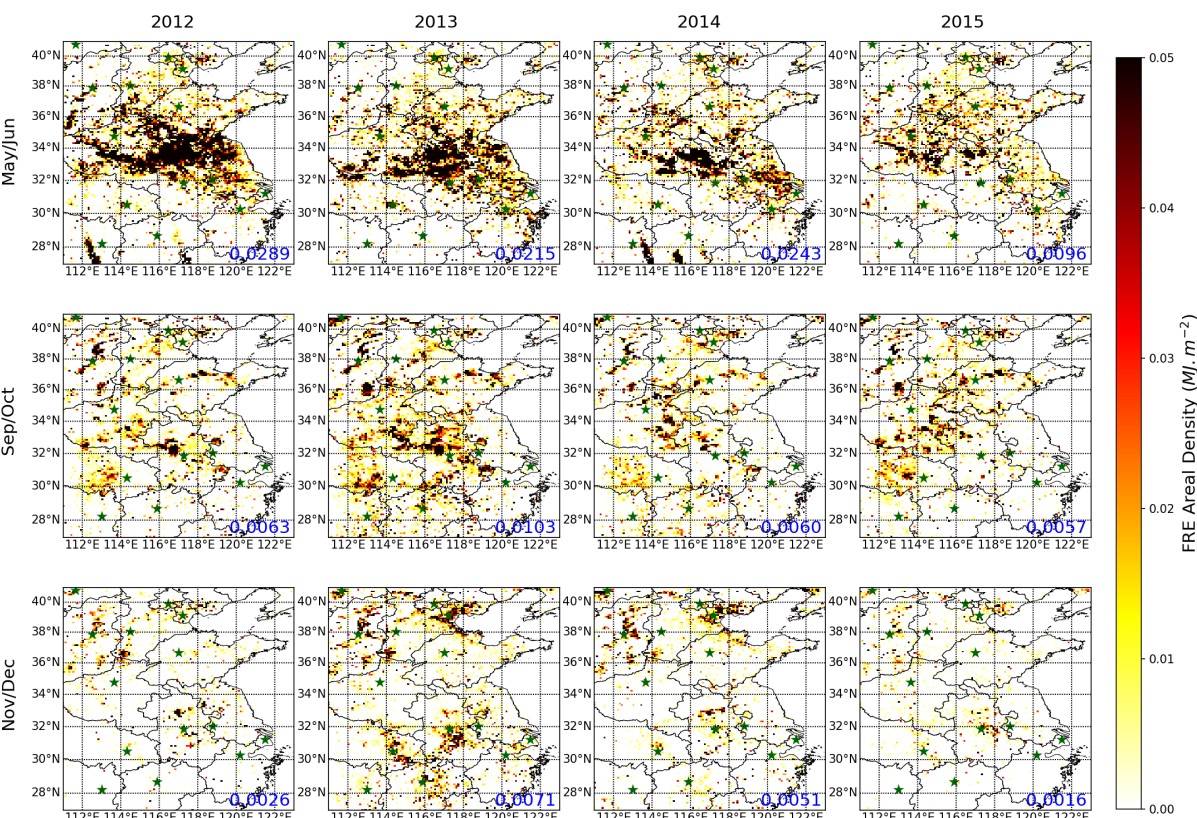

**Figure 6**: Spatial distribution of FRE areal density (MJ.m$^{-2}$, 0.1 deg grid cells) for agricultural fires in eastern China from 2012 to 2015 (top to bottom rows) split by fire season: summer (May-June, top row), autumn (Sep-Oct, middle row) and winter (Nov-Dec, bottom row). Mean regional FRE for each season is indicated in blue text, and the capital city location of each province is shown as a green star on each map. The basic layer of country/province borders within this map was created using Python Basemap librabry.





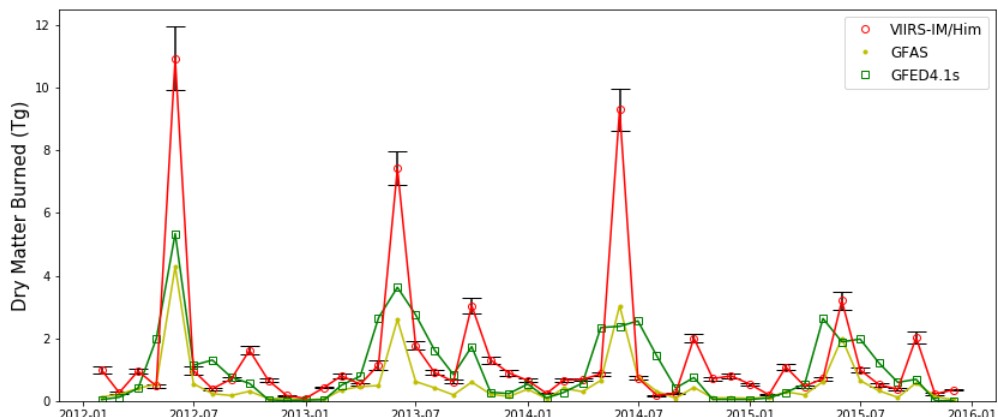

**Figure 7:** Monthly (2012-2015) time-series of total dry matter burned (DMB) retrieved using the VIIRS-IM/Him FRP product developed in this study (with standard deviation shown as black error bars), along with comparable GFAS and GFED4.1s DMB totals.

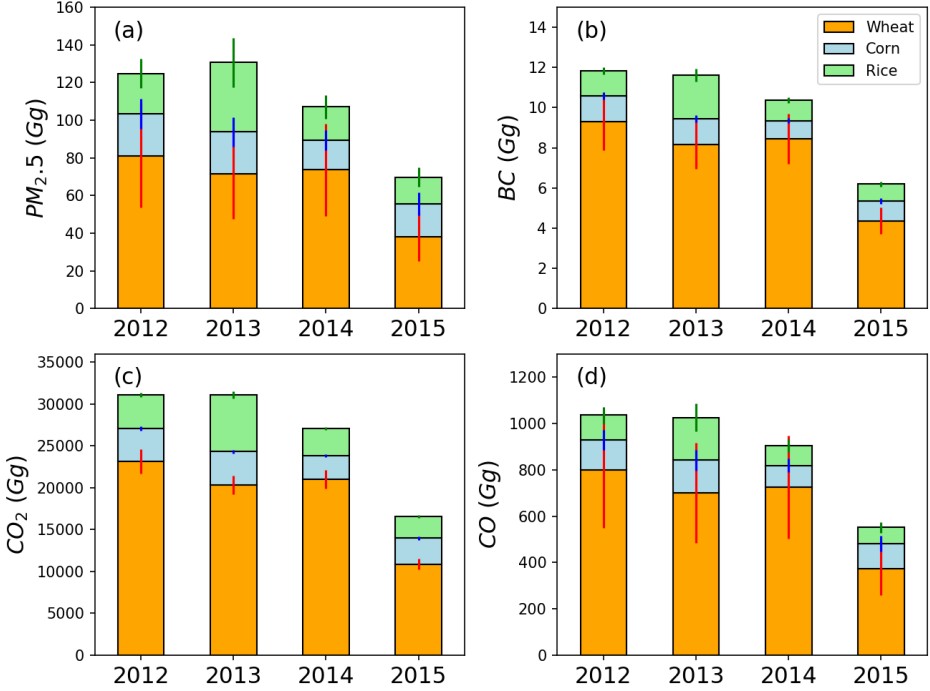

**Figure 8:** Annual total $PM_{2.5}$, BC, $CO_2$, and CO emissions for eastern China for the three main crop residues burning types (wheat, corn, rice) calculated for 2012-2015 using the VIIRS-IM/Him based emissions inventory developed herein. Coloured error bars indicate 1 standard deviation.





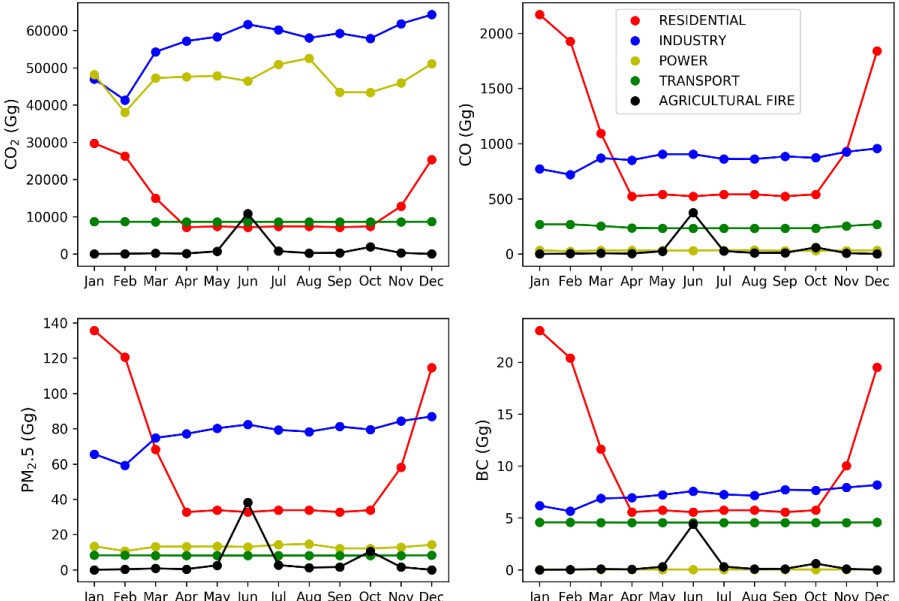

**Figure 9:** Comparison of monthly $CO_2$, CO, $PM_{2.5}$ and BC emissions from agricultural fires with those from other emission sources (residential, industry, power, transport, data source: Li et al., 2015) in the intensive burning area (32-36º N, 112-122º E) of eastern China in the year 2013.

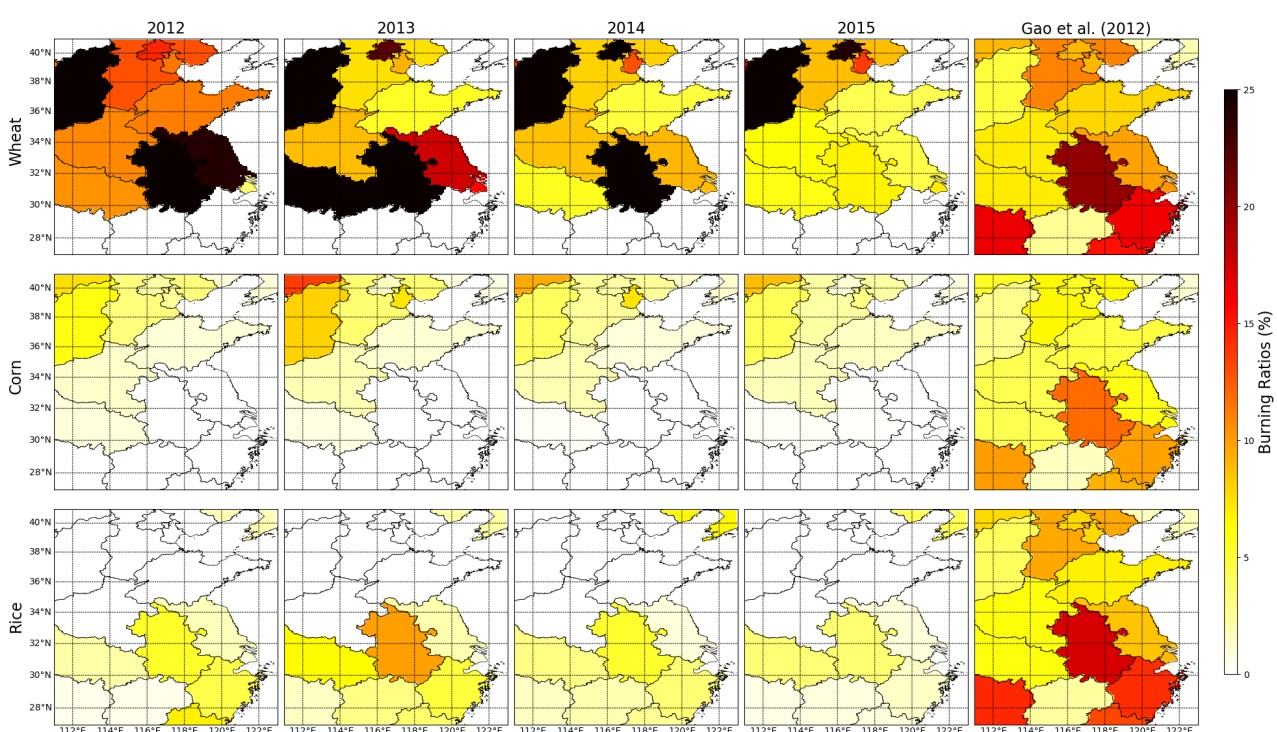

**Figure 10:** Temporal and spatial variability of province-specific percentages of crop residues burned in the fields (burning ratio metrics) of eastern China. Data are calculated using crop yield estimates from National Bureau of Statistics of China and the dry matter burned totals derived herein using our VIIRS-IM/Him DMB datasets from 2012-2015, and compared to the temporally invariant estimates provided by Gao et al., (2002, final column). The basic layer of country/province borders within this map was created using Python Basemap librabry.





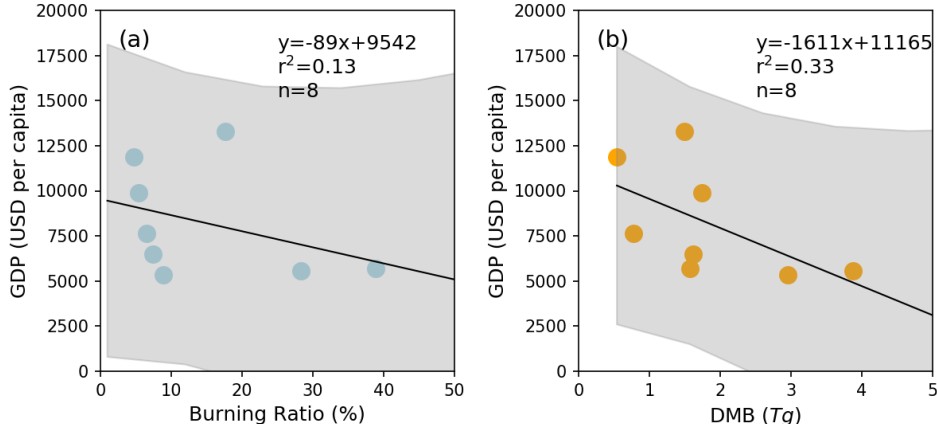

**Figure 11:** Direct comparisons of mean GDP per capita with (a) burning ratio for wheat from 2012, (b) province-specific yearly dry matter burned (DMB). The best fit linear relationships are shown, along with its equation, and the grey shaded area represents the 95% confidence limit on the relationship.

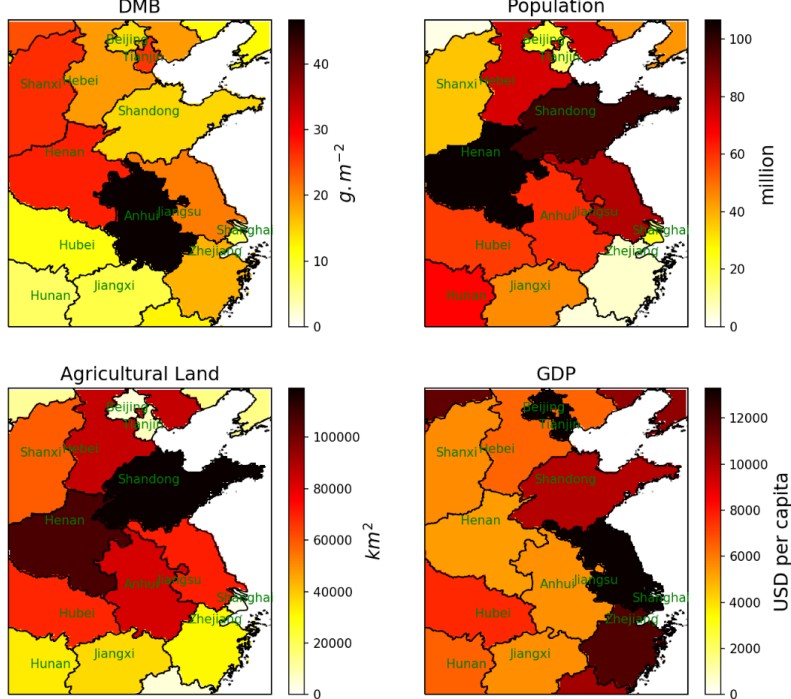

**Figure 12:** Spatial distribution of province-specific: (a) mean annual dry matter burned as calculated using the VIIRS-IM/Him approach developed herein, (b) population (Data source: Fu *et al.,* 2014a), (c) agricultural land area (Data source: GlobeLand30, http://www.globallandcover.com/) and (d) mean GDP per capita (Data source: Fu *et al.,* 2014b). The basic layer of country/province borders within this map was created using Python Basemap librabry.





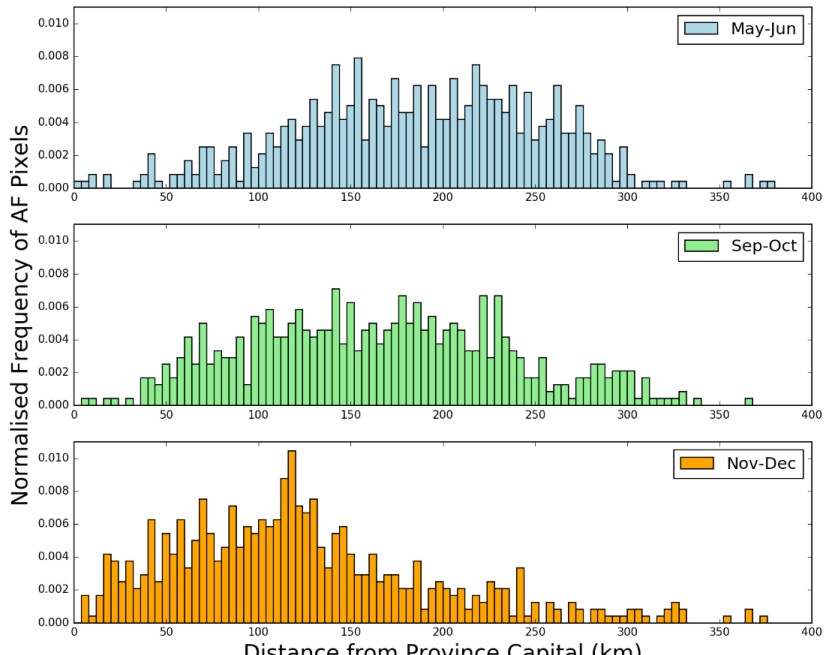

**Figure 13:** Normalised frequency distribution of distance from province capital of the top 10% of high FRE VIIRS-IM/Him product 0.1 degree grid cells during the three burning seasons: Summer - May to June (top, blue), Autumn – September to October (middle, green), and Winter - November to December (bottom, orange). A clear shift towards the origin can be observed in the Nov-Dec period compared with Sep-Oct.