# Peer review of "New eastern China agricultural burning fire emission inventory"

_Atmospheric Chemistry and Physics, 2019_

## Referee Comment (RC1) · Anonymous Referee #1 · 19 Feb 2020

General Comments

This manuscript describes the application of geostationary and polar orbiting active fire datasets to characterise agricultural burning in eastern China and to estimate the fuel consumption\trace gas emissions using fire radiative power retrievals. This study integrates VIIRS FRP retrievals with diurnal fire cycle information from the geostationary Himawari sensor to develop a fire emissions inventory optimised for agricultural burning. The improved detection of small and low intensity agricultural fires from VIIRS results in emissions estimates that are greater than those from other Earth Observation

(EO)-based emissions inventories.

The manuscript is well written and covers a topic which is suited to publication in this journal. Below are some specific and technical comments.

Specific comments

**203 – it would be useful to outline the impact of the correction of gridcell FRP for cloud cover (i.e. the percentage FRP adjustment).**

**225 and Figure 4 – the model fits the observed Himawari FRP well for most of the diurnal cycle although there is a reasonably strong secondary peak in fire activity around 20:00 which is not modelled. What is the impact of omitting the FRP contribution of this secondary peak to the daily FRE? (i.e. the difference between the 'modelled' Himawari FRP and the observed Himawari FRP). Figure 4 shows the 'summer' diurnal cycle. Are the observed Himawari diurnal cycles similar in shape in different seasons?**

**302-310 – is the winter burning season (shown in Figure 5) detected in other emissions inventories (e.g. GFED and GFAS)? It would be useful to highlight the winter burning season in Figure 7. In relation to Figure 5, is there any difference in the proportion of day/night fire detections during the winter months?**

**~321 –What might be the cause in the reduction of amount of wheat residue burnt? The wheat yield in 2015 is marginally higher than it is in previous years (Table S1)?**

**354-359 – how do the agricultural emissions derived using this approach compare with those from Li et al., 2015**

**459 – what are the combustion completeness values used in EO-derived emissions inventories such as GFED for residue burning?**

**482 – Are the DMB estimates for all crop types and were these calculated using the GlobalLand30 agricultural area estimates? How do these estimates compare to those from other studies?**

[Figure]

Technical corrections

**56 – 'this leads'**

**63 – define MODIS**

**65 – 'most BA'**

**66- define GFED**

**75 - define VIIRS**

**210 – replace 'observed' with averaged**

**223 – replace 'height' with magnitude**

**237 – replace 'see below' with Equation 9**

**248 and 252 – replace 'calculated' with estimated**

**466 – replace 'most later researchers' with 'more recent research'**

Figures

Figure 3 : Perhaps plot all of the data on the same graph and plot the data from the same day as that used in Figure 2.

Figure 4 : Is the FRP diurnal cycle from all fires in Eastern China or just agricultural fires in the region?

Figure 6 (and others): The density of map gridlines make it difficult to interpret the maps.

Figure 8 – y-axis PM2.5 subscript

[Figure]

---

## Referee Comment (RC2) · Anonymous Referee #2 · 6 Apr 2020

This manuscript could be published before addressing the comments listed following. (1) The authors declared that they could capture the small crop fires well happened in Eastern China, however, as we know, the fire size is often less than 100 by 100 square meters. They aggregated the fire data to 0.1° resolution, which is too large and not comparable with the actually existing fires. The question on small fires seem not be addressed in this manuscript. (2) Please compare your results with those from the inversions modeling or the forward simulations to check if your data are reliable. E.g., Table 2 in Cao et al. (Atmos. Chem. Phys., 18, 15017–15046, 2018), Li et al.

[Figure]

(ATMOSPHERIC ENVIRONMENT, 92, 442-448, 2014).

---

## Author Comment (AC1) · 6 May 2020

We would like to thank the referee for his/her careful and thorough reading the manuscript and consider it is well written and relevant to ACP. Below are our responses to specific and technical comments.

**203 – it would be useful to outline the impact of the correction of grid cell FRP for cloud cover (i.e. the percentage FRP adjustment).**

[Figure]

**Response:**We added following description:

"*Cloud cover (CC) fractions in some grid cells occasionally reach 0.5 (50% ), but most are zero. After the cloud cover adjustment the mean FRP areal density across the study area increased by 11.5% , so the overall effect of the CC adjustment is relatively minor.*"

**225 and Figure 4 – the model fits the observed Himawari FRP well for most of the diurnal cycle although there is a reasonably strong secondary peak in fire activity around 20:00 which is not modelled. What is the impact of omitting the FRP contribution of this secondary peak to the daily FRE? (i.e. the difference between the 'modelled' Himawari FRP and the observed Himawari FRP). Figure 4 shows the 'summer' diurnal cycle. Are the observed Himawari diurnal cycles similar in shape in different seasons?**

**Response**: The reason we do not model the secondary peak in daily FRE is that there is no satellite data from VIIRS available at this time of day to influence the peak magnitude. Instead of explicitly including this peak in the modelled diurnal cycle we include an FRP baseline above a zero value that is designed to make the daily FRE the same as if the secondary peak was modelled. This "baseline" methodology follows that of Andela et al. 2015 who used it for the same reason. To address the reviewers question we designed two simulations to compare this approach (Simulation 1) to that when the secondary peak is included (Simulation 2). In Simulation 1, the FRP derived from Himawari-8 at the VIIRS daytime and nighttime overpass times are used as $\rho$ _peak and $\rho$ _basein, whilst in Simulation 2 the distribution shown in Fig. 4 in our manuscript (red dots) is described as the sum of two Gaussian functions:

$$\widetilde{\rho}S2\left(t\right) = \sum \rho_{peaki} e^{-\frac{\left(h_t - h_{peaki}\right)^2}{2\sigma_i^2}}$$

Where $\sigma_i$ from each peak i in Fig. 4 (2.39$\pm$ 0.053 for $\sigma_1$ and 1.24$\pm$ 0.12 for $\sigma_2$ during

[Figure]

June, 1.63$\pm$ 0.041 for $\sigma_1$ and 0.60$\pm$ 0.077 for $\sigma_2$ during October) are used here, $h_{peaki}$ (h) is the hour in day when FRP reaches maximum for each of the peaks in the diurnal cycle (14.0 for $h_{peak1}$ and 21.2 for $h_{peak2}$ during June, 14.2 for $h_{peak1}$ and 18.4 for $h_{peak2}$ during October). The $\rho_{peaki}$ are the daily Himawari-8 FRP observations at those two peak maximum times.

Results from these two simulations are shown in the Figure below, which has two time series covering 10th June to 15th June each. The upper time-series shows a comparison of the two simulations using the original FRP data from Himawari-8 and Simulation 1 (S1). S1 shows a slightly overestimated baseline on 10 June and underestimation of FRP near the second peak on 13 June. Meanwhile the lower timeseries is for Simulation 2 (S2), which shows better agreement with the original Himawari-8 FRP data on 13 June but a very slight overestimation on 11 June. However, the main purpose of including the diurnal cycle is to generate correct FRE daily values, so it is better to compare FRE totals from S1 and S2 rather than hourly FRP, and we do this in the Figure 1 of this comment below.

The summed daily FRP is here used to represent FRE (without the full temporal integration, Figure 2 of this comment). The scatterplots show a direct comparison of the summed daily FRP totals from S1 (blue, left) and S2 (green, right) as compared to those from Himawari-8. Comparisons are done for Summer (June) and Autumn (Oct). The slopes of the linear best fit to these data are 1.06 and 1.15 for S1 and S2 in June, and 0.97 and 0.94 in October, suggesting that S1 performs better in both June and October. The absolute differences of S1 and S2 compared to the "true" Himawari values are however always within 10% of each other. Therefore the impact of not including the 2nd diurnal cycle peak but representing this by a baseline instead is not considered highly significant.

The observed Himawari diurnal cycles look indeed similar in shape in different seasons except for the autumn main peak is smaller and second peak time is earlier (Figure 3 of this comment ).

**302-310 – is the winter burning season (shown in Figure 5) detected in other emissions inventories (e.g. GFED and GFAS)? It would be useful to highlight the winter burning season in Figure 7. In relation to Figure 5, is there any difference in the proportion of day/night fire detections during the winter months?**

**Response**: We have enhanced Figure 7 as the reviewer suggested, and we also demonstrate that this winter burning season was not detected by either GFED or GFAS.

We also investigate the day/night fire detections during summer/autumn/winter seasons and haven't observed significant difference among them. Figure 4 of this comment gives an example from year 2013.

**∼ 321 –What might be the cause in the reduction of amount of wheat residue burnt? The wheat yield in 2015 is marginally higher than it is in previous years (Table S1)?**

**Response**: The authors believe that the most likely cause of the reduction in wheat residue burnt in 2015 compared to the prior two years is the introduction of a more aggressive policy with regards to banning agricultural residue burning. This was introduced by the local government in 2014 and was seen by us during fieldwork conducted in June 2014 and October 2015, with the latter seeing more restrictions and less burning. We also investigated yearly total FRP from MODIS Aqua in the 2003-2018 period in 30 provinces/cities (Figure 5 of this comment ). We notice that most of the provinces and cities also show this pattern of a significant reduction in burning from 2015.

**354-359 – how do the agricultural emissions derived using this approach compare with those from Li et al., 2015**

**Response**: The authors apologise here we used wrong citation in the manuscript, it should be the MIX inventory paper as below:

*Li, M., Zhang, Q., Kurokawa, J.I., Woo, J.H., He, K.B., Lu, Z., Ohara, T., Song, Y., Streets, D.G., Carmichael, G.R. and Cheng, Y.F., 2015. MIX: a mosaic Asian anthropogenic emission inventory for the MICS-Asia and the HTAP projects. Atmos. Chem. Phys. Discuss, 15(23), pp.34813-34869.*

In this paper, the authors stated that 'open biomass burning was considered as a natural emission source and excluded in the MIX inventory'. Therefore, we can only compare our emission to the listed four anthropogenic emissions in this study.

**459 – what are the combustion completeness values used in EO-derived emissions inventories such as GFED for residue burning?**

**Response**: Leeuwen et al. (2014) was the source of combustion completeness (CC) values used within GFED. It reports that 'for crop residue CC, values ranged from 65 % for cotton and sugarcane and 85 % for wheat and bluegrass'. We use a CC value to convert our fuel consumption estimates into an estimate of the amount of dry matter that is actually set fire to in the fields. We assume a CC of what of 86% (Table S2) based on Huang et al., 2012, which is very close to the 85% assumed in GFED. So therefore our calculated "residue amount" is given by (fuel mass burned/0.86). This then is compared to the wheat yield data to give our "burning ratios" presented in Figure 10.

**482 – Are the DMB estimates for all crop types and were these calculated using the GlobalLand30 agricultural area estimates? How do these estimates compare to those from other studies?**

**Response**: Yes, the DMB estimates in this study are for all crop types, and they were calculated using the GlobalLand30 landcover map for agricultural areas. We used the MIRCA2000 rotation cultivation dataset to identify which crop type was burning at a particular location at different times of year (Figure S1).

[Figure]

Figure 7 gives comparison of DMB reported in this paper compared to that of GFAS and GFED. We have generally higher estimates than GFAS/GFED thanks to the ability of the VIIRS sensor to identify far lower FRP fires (Zhang et al., 2017). Since agricultural residue fires are typically quite small and of low intensity, this ability significantly improves the overall estimate of DMB for these types of fires. Most regional crop residue burning estimates are based on the aforementioned "bottom up" crop yield-based approach and whilst they often do not report DMB estimates they do report CO2 emission estimates which are directly proportional to DMB because CO2 represents almost 95% of the carbon released. We already compare the CO2 emissions values from our methodology to those from the "bottom up" approach in Table 2.

**56 – 'this leads'**

**Response:**Revised as suggested.

**63 – define MODIS**

**Response:**Revised as suggested.

**65 – 'most BA'**

**Response:**Revised as suggested.

**66- define GFED**

**Response:**Revised as suggested.

**75 - define VIIRS**

**Response:**Revised as suggested.

**210 – replace 'observed' with averaged**

**Response:**Revised as suggested.

**223 – replace 'height' with magnitude**

**Response:**Revised as suggested.

**237 – replace 'see below' with Equation 9**

**Response:**Revised as suggested.

**248 and 252 – replace 'calculated' with estimated**

**Response:**Revised as suggested.

**466 – replace 'most later researchers' with 'more recent research'**

**Response:**Revised as suggested.

**Figures**

**Figure 3 : Perhaps plot all of the data on the same graph and plot the data from the same day as that used in Figure 2.**

**Response:**We have edited the plot as the referee suggests. Though the data from Himwari-8 is not available on the day used for this Figure 2 as the satellite was only launched some years later.

**Figure 4 : Is the FRP diurnal cycle from all fires in Eastern China or just agricultural fires in the region?**

**Response:**Agricultural fire is the dominant biomass burning in Eastern China, especially during burning season (accounts for over 99% of total FRP). We did include all the fires when calculating the FRP diurnal cycle. However, when we excluding those non-agricultural fires, the change to diurnal cycle is very limited. (Figure 6 of this comment ). We got almost similar summer diurnal cycle $\sigma$ value (2.40) compare to the one we use in this paper (2.39).

**Figure 6 (and others): The density of map gridlines make it difficult to interpret the maps.**

**Response:**We removed the gridlines and changed color theme to make the maps more readable.

**Figure 8 – y-axis PM2.5 subscript**

**Response:**Revised as suggested.

———————————————

Hourly FRP (MW)

Himawari-8

Himawari-8

**Fig. 1.**

[Figure]

**Summer**

**Autumn**

**Fig. 2.**

[Figure]

[Figure]

**Fig. 3.**

[Figure]

**Fig. 4.**

[Figure]

**Fig. 5.**

Fig. 6.

---

## Author Comment (AC2) · 6 May 2020

Response to Referee # 2

**Response:** The referee has raised some concern about the issue of data aggregation to 0.1-degree resolution, which we believe we have dealt with very carefully in the paper. Below are our responses to each detailed comment provided by the referee:

[Figure]

**(1) The authors declared that they could capture the small crop fires well happened in Eastern China, however, as we know, the fire size is often less than 100 by 100 square meters. They aggregated the fire data to 0.1-degree resolution, which is too large and not comparable with the actually existing fires. The question on small fires seem not be addressed in this manuscript.**

**Response:**

The authors are fully aware that the Chinese agricultural lands are small and that the residue fires are also therefore small, often far less than the 100 100 m as the reviewer suggests. This is the reason we are using VIIRS-IM FRP data, which is based on 375 m pixels, rather than MODIS with its 1 km pixels. Active fire detection algorithms can identify fires covering only 0.0001 of a pixel area, and the smaller VIIRS pixels thus enable us to detect fires down to around 5 $m^2$ at night and perhaps down to around twice that by day (see Zhang et al., 2017 for details). Below we show a figure with VIIRS and MODIS fire pixel footprint sizes overlain on Google Earth for 11[th] June 2015 – and this highlights the advantage of the smaller pixel area of VIIRS. What we are doing is detecting the active fires at the full resolution of VIIRS, thus enabling us to capture even the small fires, and then aggregating the FRP from all of these fires detected in each 0.1-degree grid cell. So each grid cell represents the total FRP coming from all fires detected within it at the particular overpass time.

**(2) Please compare your results with those from the inversions modelling or the forward simulations to check if your data are reliable. E.g., Table 2 in Cao et al. (Atmos. Chem. Phys., 18, 15017–15046, 2018), Li et al.(ATMOSPHERIC ENVIRONMENT, 92, 442-448, 2014).**

The authors are struggling to compare our inventory data to Cao's et al 2018 modelling results or Li's et al. 2014 atmospheric species' concentration results. Below summarises our best effort comparison:

The measurements of NMVOC emission factors for different crop residues in China was not target for this study. To estimate NMVOC emissions, first we found GFAS uses a generic emission factor of 9.9g/kg for NMHC emitted from agricultural fires. When applying this to our data, we got an estimated yearly NMHC emission in Eastern China of 106-188 Gg in 2012-2015. Jain et al., 2014 suggested that the total emitted NMVOC from India is around 1.46 Mt while total NMHC is around 0.65 Mt. We can get a rough ratio of 2.25 for NMVOC/NMHC. If we assume Eastern China contribute a quarter of total agricultural burning to whole China, according to the publications we cited in Table 2, the total NMVOC emission is 0.96-1.69 Tg, lower but comparable to Cao et al., 2018 results of 2.08 to 3.13 (average 2.48) Tg yr$^{-1}$ from biomass burning. It is also following our comparison in CO2 that our values of emissions are generally smaller than results using CYBA (Crop Yield Based Approaches) method.

Li et al. 2014 only reported concentrations rather than emissions, making it even more difficult to compare. The only thing we can try here is to compare the ratio of BC/PM2.5.The average PM2.5 concentration was reported 110.7 mg/m3, containing 7.3 mg/m3 EC in their study, which accounts around 6.5% of the particle. Our yearly emission results show that around 9% particulate mass is around black carbon, slightly higher but reasonably close to Li's results. This could be because we collected our samples close to the fire, limiting the impact of aerosol aging during transportation and consequently secondary organic aerosol formation.
* * *
[Figure]

**Fig. 1.**